# Comparison of Various Extraction Approaches for Optimized Preparation of Intracellular Metabolites from Human Mesenchymal Stem Cells and Fibroblasts for NMR-Based Study

**DOI:** 10.3390/metabo14050268

**Published:** 2024-05-07

**Authors:** Slavomíra Nováková, Eva Baranovičová, Zuzana Hatoková, Gábor Beke, Janka Pálešová, Romana Záhumenská, Bibiána Baďurová, Mária Janíčková, Ján Strnádel, Erika Halašová, Henrieta Škovierová

**Affiliations:** 1Biomedical Centre Martin, Jessenius Faculty of Medicine in Martin, Comenius University in Bratislava (JFM CU), Malá Hora 4C, 036 01 Martin, Slovakia; slavomira.novakova@uniba.sk (S.N.); zuzana.hatokova@uniba.sk (Z.H.); janka.palesova@uniba.sk (J.P.); romana.zahumenska@uniba.sk (R.Z.); jan.strnadel@uniba.sk (J.S.); erika.halasova@uniba.sk (E.H.); henrieta.skovierova@uniba.sk (H.Š.); 2Institute of Molecular Biology, Slovak Academy of Sciences, Dúbravská Cesta 21, 845 51 Bratislava, Slovakia; gabor.beke@savba.sk; 3Department of Stomatology and Maxillofacial Surgery, University Hospital in Martin and JFM CU, Kollárova 2, 036 01 Martin, Slovakia; maria.janickova@uniba.sk

**Keywords:** mesenchymal stem cells, intracellular metabolites, NMR

## Abstract

Metabolomics has proven to be a sensitive tool for monitoring biochemical processes in cell culture. It enables multi-analysis, clarifying the correlation between numerous metabolic pathways. Together with other analysis, it thus provides a global view of a cell’s physiological state. A comprehensive analysis of molecular changes is also required in the case of mesenchymal stem cells (MSCs), which currently represent an essential portion of cells used in regenerative medicine. Reproducibility and correct measurement are closely connected to careful metabolite extraction, and sample preparation is always a critical point. Our study aimed to compare the efficiencies of four harvesting and six extraction methods. Several organic reagents (methanol, ethanol, acetonitrile, methanol–chloroform, MTBE) and harvesting approaches (trypsinization vs. scraping) were tested. We used untargeted nuclear magnetic resonance spectroscopy (NMR) to determine the most efficient method for the extraction of metabolites from human adherent cells, specifically human dermal fibroblasts adult (HDFa) and dental pulp stem cells (DPSCs). A comprehensive dataset of 29 identified and quantified metabolites were determined to possess statistically significant differences in the abundances of several metabolites when the cells were detached mechanically to organic solvent compared to when applying enzymes mainly in the classes of amino acids and peptides for both types of cells. Direct scraping to organic solvent is a method that yields higher abundances of determined metabolites. Extraction with the use of different polar reagents, 50% and 80% methanol, or acetonitrile, mostly showed the same quality. For both HDFa and DPSC cells, the MTBE method, methanol–chloroform, and 80% ethanol extractions showed higher extraction efficiency for the most identified and quantified metabolites Thus, preparation procedures provided a cell sample processing protocol that focuses on maximizing extraction yield. Our approach may be useful for large-scale comparative metabolomic studies of human mesenchymal stem cell samples.

## 1. Introduction

One of the approaches that enables a comprehensive view of the events in cells is metabolomics. Analyzing a broad spectrum of metabolites provides specific information about processes under physiological and pathological conditions. Reproducible analytical techniques, characterized by sufficiently high sensitivity, wide range, and high resolution, are suitable for studying metabolic profiles [1]. One method involves quantitative nuclear magnetic resonance spectroscopy (NMR), which is also used in our study. In metabolomics, LC-MS, GC-MS, and NMR spectroscopy are three widely used analytical techniques. Although LC-MS and GC-MS remain the most popular, constituting over 80% of published metabolomic studies, NMR-based approaches have seen growing use. NMR spectroscopy has several advantages when compared to LC-MS and GC-MS platforms. Specifically, NMR spectroscopy offers non-destructive, unbiased, easily quantifiable, and minimally demanding processes regarding chromatographic separation, sample treatment, or chemical derivatization. Beyond these merits, NMR excels in detecting and characterizing compounds that present challenges for LC-MS analysis, including sugars, organic acids, alcohols, polyols, and other highly polar substances [2,3,4,5,6]. Despite the constant technical improvements of instruments, sample preparation is essential for accurate measurement to achieve accurate and reproducible results. The extraction of intracellular metabolites is a critical step in metabolomic analysis that involves different harvesting approaches and organic solvents to achieve optimal metabolite recovery and preservation [7,8,9,10,11,12]. Although the pre-analytical phase greatly influences the results, few studies have been devoted to it. None of the methods are universal; in connection with the scientific focus of our working group, we focused on the isolation of intracellular metabolites from mesenchymal stem cells. Specifically, we used dental pulp stem cells (DPSCs) and adult human dermal fibroblasts (HDFa). HDFa are morphologically indistinguishable from MSCs, and their properties meet the characteristics of MSCs according to ISCT (International Society for Cell and Gene Therapy) [13]. It has been hypothesized that fibroblasts are essentially senescent MSC cells of the same cell type [14]. The clinical application of stem cells is the subject of intense debate. The biological mechanisms of MSC action still need to be fully elucidated [15,16]. Metabolomics is one of the methodologies that can help clarify these mechanisms.

Thus, proper intracellular metabolite isolation techniques are essential for collecting cells from a culture, with subsequent disruption of the cellular matrix to release intracellular metabolites. Cell extraction plays a crucial role in metabolomic analysis for several reasons. It ensures the isolation of metabolites from the cellular environment, reducing potential interference from extracellular compounds. In addition, it provides an opportunity to stop cellular metabolism rapidly, preserving metabolite composition when harvesting. Finally, it enables the extraction of metabolites from different cellular compartments, such as the cytoplasm, nucleus, mitochondria, or other organelles, providing insight into compartment-specific metabolism. Isolation involves cell harvesting, metabolism quenching, cell disruption, and the extraction of metabolites [9,17]. These steps can be carried out in several ways, ultimately affecting the measurement results. For adherent cells, detachment from the culture flask surface is the initial step in cell harvesting. Techniques such as trypsinization, scraping, or the application of cell dissociation buffers can be used. Care should be taken to minimize cell stress and damage during detachment to avoid potential alterations in metabolite composition. Trypsin is associated with metabolite leaking and affects the metabolite expression rate [7,10,18,19]. Organic solvents play a crucial role in metabolite extraction, as they facilitate the release of intracellular metabolites from the cellular matrix. Commonly used organic solvents include methanol, ethanol, acetonitrile, chloroform, and their mixtures. Each solvent has advantages and disadvantages regarding metabolite extraction efficiency, solubility, and compatibility with downstream analytical techniques [8,12].

Our work compared trypsinization and the method of direct scraping into organic solvent and extraction using several organic reagents with different polarities. We applied the technique of metabolite extraction with simultaneous protein precipitation. This method allows for a quick and efficient obtaining of metabolites and proteins in one step, thus enabling the correct normalization of the measured relative abundances [20,21]. One can fully understand the dynamics of intracellular metabolites in cellular physiology and disease by integrating analytical techniques, sample preparation methods, and data analysis tools. This field holds immense promise for advancing our knowledge of cellular metabolism and translating these insights into clinical applications and biotechnological advances.

## 2. Materials and Methods

### 2.1. Cultivation of Human Adherent Cells

Commercially available cells, human dermal fibroblasts (HDFas, Gibco, Billings, MT, USA) and dental pulp stem cells (DPSCs, Lonza, Basel, Switzerland), were used in the work. The cells were cultured in DMEM: F12 + GlutaMAX (Gibco) commercial culture medium supplemented with 10% fetal bovine serum (FBS, Biosera, Cholet, France) and antibiotics (100 U/mL penicillin and 100 µg/mL streptomycin, Biosera). The cells were grown in a humid atmosphere at 37 °C and 5% CO_2_. After reaching an 80–90% confluence, the cells were detached using TrypLE Express Enzyme (Gibco) and seeded at a 5000 cells/cm^2^ density. In this study, cells from passage no. 4 were used. Over the entire course of the experiment, the morphology of the cells was monitored by light microscopy (Optica).

### 2.2. Isolation of Intracellular Metabolites for Harvesting Method Optimization

On the day that the cells reached an 80% confluence, the HDFas and DPSCs were washed twice with Dulbecco’s PBS solution (DPBS, Gibco) that was prewarmed (warm DPBS, 37°) or cooled on ice (cold DPBS, 4 °C). Then, the cells were scraped from the culture flask using an extractant of 50% (*v*/*v*) methanol. Other cells were trypsinized using TrypLE Express Enzyme (Gibco) or 0.25% trypsin-0.53 mM EDTA (Sigma, Oakville, ON, Canada). After detaching, the cells were resuspended in a 50% methanol solution. Then, the cell lysates were transferred to microtubes (Eppendorf), sonicated 3 × 10 s, and incubated for 20 min at −20 °C. Subsequently, the samples were spun at 14,000× *g* and 4 °C. The supernatant was stored at −80 °C until use. The sediment with precipitated proteins was resuspended in SDT buffer (4% SDS, 100 mM DTT, 100 mM Tris-HCl pH 7.4) with shaking at 20 °C and centrifuged at 14,000× *g* and 20 °C. The supernatant was stored at −80 °C until use.

### 2.3. Isolation of Intracellular Metabolites Using a One-Phase System

The cells reached an 80% confluence, and the HDFas and DPSCs were washed twice with Dulbecco’s PBS (4 °C). Then, the cells were scraped from the culture flask using an appropriate extractant (50% (*v*/*v*) methanol, 80% (*v*/*v*) methanol, 70% (*v*/*v*) acetonitrile, or 80% (*v*/*v*) ethanol). Then, the cell lysates were transferred to microtubes (Eppendorf), sonicated 3 × 10 s, and incubated for 20 min at −20 °C. Subsequently, the samples were spun at 14,000× *g* and 4 °C. The supernatant was stored at −80 °C until use. The sediment with precipitated proteins was resuspended in SDT buffer (4% SDS, 100 mM DTT, 100 mM Tris-HCl pH 7.4) with shaking at 20 °C and centrifuged at 14,000× *g* and 20 °C. The supernatant was stored at −80 °C until use.

### 2.4. Isolation of Intracellular Metabolites Using a Two-Phase System

When the density of the cells reached 80%, the HDFas and DPSCs were washed with cold (4 °C) DPBS solution (Gibco) and scraped from the bottom of the culture bottle using 75% methanol. Protocols were performed according to Lorentz et al., 2011 with minor modification [8] (methanol–chloroform extraction) and Luo et al., 2020 [19] (methyl-t-butyl ether (MTBE) extraction). Briefly, the cell lysates were transferred to a microtube, where chloroform (in ratio 9:1, methanol: chloroform) or MTBE were subsequently added. In the case of the methanol–chloroform extraction, samples were sonicated (3 × 10 s), incubated for 20 min at 4 °C with agitation, and spun at 14,000× *g* and 4 °C. The polar and hydrophobic phases were stored at −80 °C until further use. The interphase that contained the proteins was resuspended in SDT buffer, sonicated (3 × 10 s), and spun at 14,000× *g* and 20 °C. The supernatant was stored at −80 °C until further use. In the case of the MTBE extraction, the samples were sonicated (3 × 10 s), and then MTBE in ratio 1:3 (methanol: MTBE) was added to the cell lysate and incubated for 6 min at room temperature with agitation. Then, water was added and the samples were agitated for 2 min more at room temperature, sat down for 10 min, and spun at 14,000× *g* and 4 °C. Next, the samples were processed as described in the methanol–chloroform extraction.

### 2.5. Determination of Protein Concentration and Verification of Their Quality

The concentration of soluble proteins was determined with a Pierce 660 nm kit (Thermo Fisher Scientific, Waltham, MA, USA) according to the manufacturer’s instructions. The quality of the proteins was verified by their electrophoretic separation in a polyacrylamide gel under denaturing conditions (PAGE). Then, 25–30 µg of protein were loaded on a 12% Tris–glycine gel. Electrophoresis was performed in Tris–glycine electrophoresis buffer (Serva). The gel was stained with Colloidal Coomassie Blue and washed in distilled water overnight.

### 2.6. NMR Analysis

#### 2.6.1. Sample Preparation before Measurement

The supernatant was dried in a SpeedVac vacuum dryer (ThermoScientific, Waltham, MA, USA). The dry matter was dissolved in 550 µL of phosphate-buffered deuterated water (200 mM, pH 7.4, which contained 0.2 mM TMS-d_4_ (trimethylsilylpropionic acid-d_4_) as a chemical shift reference and was assigned a chemical shift of 0.000 ppm during data processing).

#### 2.6.2. NMR Measurement

NMR data were acquired using a 600 MHz Avance III NMR spectrometer (Bruker, Munich, Germany) equipped with a cryoprobe at an acquisition temperature of T = 310 K. The samples were freshly prepared and tempered for 5 min at 310 K before measurement. 1D and 2D NMR spectra were measured for each sample. Standard metabolomic profiling protocols from Bruker were modified as follows: (i) NOESY with pre-saturation (noesygppr1d): FID size 64k, dummy scans: 4, number of scans: 128, spectral width: 20.4750 ppm; (ii) cpmg with pre-saturation: FID size 4k, dummy scans: 8, number of scans: 512, spectral width: 16.0125 ppm; (iii) homonuclear J-resolved spectra: FID size 8k, dummy scans: 16, number of scans: 64, COSY with pre-saturation (cosygpprqf) acquired for randomly chosen 10 samples: FID size 4k, dummy scans: 8, number of scans: 12, spectral width 16.0125 ppm. All the experiments were performed with a relaxation delay of 4 s.

#### 2.6.3. Data Analysis

NMR spectra were binned to 0.001 ppm, ranging from 0.00 to 10.00 ppm. The multiplicity of the peaks was confirmed in J-resolved spectra, and the homonuclear cross-peaks were confirmed in COSY spectra. Spectra were solved using an online metabolomics database (www.hmdb.ca, accessed 01.01–31.05/2023), a free trial version of Chenomx software (NMR suite 9.0), an in-house metabolite database, and metabolomics literature searches. Subsequently, after identifying the metabolites, subregions of the spectra were selected, to which only one metabolite was assigned or minimally influenced by other co-occurring metabolites. The integrals of the selected metabolites were calculated from the binned spectra (size 0.001 ppm). These data express the relative concentration of a particular metabolite in the sample. Metabolites showing weakly intense peaks or strongly overlapping peaks were excluded from the quantitative evaluation (Appendix A). The relative concentrations of metabolites in the samples were normalized to the amount of total protein (µg).

### 2.7. Statistical Analysis

Differentially abundant metabolites were filtered on effect size using ANOVA *p* ≤ 0.05 followed by Tukey’s post hoc test. The significance criterion was a corrected *p*-value < 0.05 (Benjamini–Hochberger correction). Heat maps were performed in R programming language (version 4.0.3) [22]. Sparse PLS-DA (Partial least squares-discriminant analysis) [23], which reduces multidimensional metabolomic data into 2D space including discriminatory steps, was performed using the online tool Metaboanalyst v6.0, accessed on 01–31.03/2024 [24].

## 3. Results

To perform the methods precisely, all the samples were analyzed in triplicate. First, we compared (i) adherent cells detached from flasks based on enzymatic assisted or scraping into an organic solvent and (ii) various organic solvents with different polarities to optimal and effective extraction. The entire experimental workflow is shown in Figure 1 to illustrate the specific design of this work.

### 3.1. Evaluation of Impact of Harvesting Method on Intracellular Metabolome

To determine the impact of mechanical vs. enzymatic detachments of samples, we analyzed four experimental approaches: direct scraping to organic solvent following previous washing with warm (37 °C) or ice cold PBS buffer (4°) (from now on referred to as cold or warm PBS) and treatment using TrypLE Express enzyme (from now on referred to as TrypLE) or 0.25% Trypsin–EDTA 0.53 mM reagent (from now on referred to as Trypsin–EDTA). Following trypsinization, the samples were also subjected to extraction with 50% (*v*/*v*) methanol with the same volume that was used in the case of direct solvent scraping. In our experiments, two human adherent cells were applied, HDFas and DPSCs, after reaching a confluence of 80% (Appendix A). In all four cases, for both types of cells, we identified a dataset of 29 polar metabolites, clustered the 12 classes, and determined their relative abundances. These were subsequently normalized to the total amount of precipitated proteins (µg). This type of normalization was identified as being fully compatible with cell count normalization [20]. The quality of the isolated proteins was detected on PAGE gels. In total, we tested six combinations to compare the impact of the harvesting method: 1. scraping/cold PBS vs. scraping/warm PBS; 2. TrypLE harvest vs. Trypsin–EDTA harvest; 3. scraping/cold PBS vs. TrypLE harvest; 4. scraping/cold PBS vs. Trypsin–EDTA harvest; 5. scraping/warm PBS vs. TrypLE harvest; and finally, 6. scraping/warm PBS vs. Trypsin–EDTA harvest. The metabolites with significant changes (corrected *p* < 0.05) according to the methods are highlighted with the darker color on the heat map (Figure 2). Also, the all metabolites are listed in Appendix A. Differences along enzymatic treatments or direct scraping into 50% (*v*/*v*) methanol preceded by washing using PBS with different temperatures exhibited similar performances for both types of cells. We determined that no statistically significant differences exist between the treatments with two types of trypsin or with the effect of temperature of washing buffer (Figure 2). Following our experimental workflow, we compared the direct scraping of cells into an organic agent (50% methanol) and trypsinization with two types of trypsin, all in the several combinations shown in the heat map (Figure 2). Our analysis did not show consistent changes in complete metabolite class, and we identified only increased/decreased abundances of individual metabolites from various groups. Nevertheless, metabolites belonging to the group of amino acids had the largest share in the recorded changes. For the HDFa cells, 13 out of the 29 metabolites showed statistically significant differences. We observed significant changes in alanine, valine, and lactate abundances for all the PBS washing buffer vs. trypsin comparisons. On the other hand, UDP-Glc, hypoxanthine, and AXP (AMP, ADP) showed differences only when comparing scraping with previously warm PBS buffer washing vs. TrypLE or Trypsin–EDTA detachment. Finally, differences in glutamine abundances were observed only in comparison to warm PBS-TrypLE conditions (Figure 2). For the DPSCs, 12 out of the 29 metabolites showed statistically significant differences. Similarly to the HDFa cells, alanine (cold PBS-EDTA, cold PBS-TrypLE, warm PBS-TrypLE), lactate (cold PBS- Trypsin–EDTA, cold PBS-TrypLE), and glutamine (cold PBS-EDTA, cold PBS-TrypLE, warm PBS-TrypLE) were affected. Contrary to the HDFas, we determined more affected amino acids, aspartate, threonine, glutamate, glutamine, phenylalanine, and leucine, and found no changes in the abundances of UDP-Glc, hypoxantine, and AXP. The differences in harvesting methods were assessed by using sPLS-DA analyses when normalized levels of the metabolites were used as input variables. The 2D visualizations suggest that harvesting using direct scraping into the organic solvent is not dependent on the PBS medium temperature, as the data from these methods were instead clustered together. On the other hand, the clusters from the metabolic data of the Trypsin–EDTA and TrypLE harvesting methods were slightly shifted away from PBS for both cell types, HDFas as well as DPSCs, slightly overlapping with each other for both types of cells (Figure 3), and were in excellent agreement with other. More detailed data are presented in this study. The results are often normalized to proteins isolated from cells growing on parallel cultured flasks. When we applied this method, normalization brought us a smaller amount (2) of statistically significant (corrected *p*-value <0.05) identified metabolites. These differences point to the importance of correct normalization.

We compared individual harvesting methods by comparing normalized abundances of particular metabolites. Cells directly scraped to 50% methanol with previous washing with cold PBS were determined as a reference set (100% of individual abundances). We determined that in the case of HDFa and DPSC cells, direct scraping to organic solvent following previous washing with cold or warm PBS showed higher abundances for the most identified and quantified metabolites (Figure 3, Appendix A). It is correct to say that due to the limited dataset (29 metabolites), we included all metabolites in the analysis shown in the heat map (Figure 2). We did not set a threshold for the coefficient of variation (% CV), which is usually limited to % CV ≤ 30–35% for LC-MS analysis. The % CV is shown in Figure 4, and it ranges depending on the cells and the harvesting or extraction methods used. For the DPSC cells, the average % CV is more balanced for all harvesting methods (22–24 out of 29 metabolites with % CV ≤ 35%), while in the case of the HDFa cells, trypsinization techniques showed a higher % CV (15 and 19 out of 29 metabolites with % CV ≤ 35%).

We observed % CV depending on the harvesting method, as Rushing et al. reported [25] (UHPLC-HRMS-based measurement). Specifically, we determined the % CV for lysine in the DPSC cells to have the following values: 27.1% for scraping with cold PBS, 34.5% for scraping with warm PBS, 14% for TrypLE use, and 7.6% for Trypsin–EDTA use. Rushing et al. [25] have reported on the same metabolite, with a % CV of 31.3% when the scrape–freeze–thaw method was used, 38.4% for scrape–homogenization, 27.9% for trypsin–homogenization, and 8.5% for a trypsin–freeze–thaw method. The entire dataset is included in the Appendix A.

### 3.2. Evaluation of Impact of Extraction Solvent on Intracellular Metabolome

To compare the efficiency of several extraction methods, we selected four extractants that differ in polarity (50% methanol, 80% methanol, 80% ethanol, 70% acetonitrile) for a one-phase system and two methods for a biphasic system (MTBE method and methanol–chloroform). The direct cell scraping into 50% methanol following previous washing with cold PBS was determined as a reference set (100% of individual abundances). Then, for both the HDFa and DPSC cells, we determined that the MTBE method, methanol–chloroform extraction, and 80% ethanol extraction showed higher extraction efficiencies for the most identified and quantified metabolites (Figure 5, Appendix A). When comparing harvesting methods, we observed the most significant changes in the groups of amino acids. When comparing different extraction approaches, we identified the most statistically significant changes in the groups of amino acids, carboxylic acids, and then purine and pyrimidine nucleotides, their sugars, and derivatives (Figure 2). In detail, in the DPSC cells, extractions involving MTBE resulted in statistically significant differences for several amino acids, uridine, hypoxanthine, AXP, and lipoproteins, compared to all others (Figure 2). Interestingly, the extractions involving MTBE resulted in the lowest extraction efficiency in comparison to all the agents in the case of AXP.

We found higher correlation coefficients for several metabolites when measured using NMR, which corresponds with the findings of Kostidis et al. [9] (NMR-based measurement). Specifically, the % CV for lysine was determined depending on the extraction solvent and cells. The average of % CV for the DPSCs across all the extraction agents used was 30.5%; for the HDFa cells the average was 15–45.7% across all the extraction agents. Kositidis reported a 36.88% CV for lysine (BHP2-7 cell line, 100% MetOH extraction). For the HDFa cells (apart from the methanol–chloroform method), 21–27 out of the 29 metabolites showed % CV ≤ 35% when using different extraction methods. For the DPSC cells, the situation was more balanced; all the extraction methods showed 21–24 out of the 29 metabolites, with % CV ≤ 35%.

## 4. Discussion

Cellular metabolism is a highly dynamic and interconnected system involving numerous biochemical reactions and pathways. Characterizing the complete set of metabolites within a cell has become increasingly important in understanding cellular physiology, disease mechanisms, or drug discovery. Intracellular metabolomics has emerged as a powerful tool for unraveling the complexity of cellular metabolism. The extraction of intracellular metabolites is a critical step in metabolomic analysis. Different harvesting approaches and organic solvents are employed to achieve optimal metabolite recovery and preservation. Compared to preparing other biological samples (urine, plasma, serum), preparing samples from human adherent cells for metabolomic studies of intracellular metabolites is more complicated. It consists of cell harvesting, metabolism quenching, cell disruption, and metabolite extraction [9,17].

Each of these steps can be carried out in several ways, which can ultimately affect the overall measurement result. For this reason, recently, more studies have focused on the development of correct procedures for preparing samples from human adherent cells for metabolomic studies [9,12,19,26].

In the first part, we compared cell collections by scraping or trypsinization and incorporated the quality of the washing solution and trypsinization agents. The choice of washing solution, quantity, and washing time can affect the quenching of cells and, ultimately, the result. Mili et al., 2020 [26] have reported that a single wash with PBS using 6 mL volume sufficiently removes the culture medium and preserves the concentration of intracellular metabolites. Several studies were carried out by washing cells with PBS or 0.9% NaCl at different temperatures [27,28]. We aimed to compare washing cells with warm (37 °C) and cold (4 °C) PBS, with a focus on the influence of quenching the cells before applying direct scraping into the organic agent. To the best of our knowledge, no research group has yet focused on this point of view.

Similarly, several types of trypsinizing agents are on the market (Accutase^TM^, EDTA, TrypLE Enzyme Express, Trypsin–EDTA), some of which are offered as a more gentle option for cells [29]. We compared two of them, porcine trypsin and an animal-free alternative TrypLE Enzyme Express. We found that neither the temperature of the washing solution nor the type of selected trypsinizing agent affected the change in the abundance of metabolites. Several studies have confirmed the effect of trypsinization on metabolite leakage and recommend direct extraction into an organic agent [7,19,20]. The impact of a harvesting method on an overall metabolome is also influenced by the dataset size.

Nevertheless, metabolites determined by NMR are abundant and play an important role in the main metabolic pathways. Also, we identified several metabolites that showed statistically significantly changes, representing crucial players. In the comparative analyses in our study, metabolites classified into the group of amino acids, peptides, and their derivatives showed the most statistically significant changes. This points to their importance, as stem cells can utilize amino acids for protein synthesis and energy production, thereby contributing to regulating the intracellular environment’s homeostasis [30]. Amino acids also provide energy during MSC differentiation [31]. Another factor that affects the obtained results is the normalization of the obtained data on relative abundances [32]. There are several possibilities, including normalization to DNA, proteins, or biomass [11,33,34]. Normalization to the total amount of proteins is increasingly coming to the fore [20,21]. The advantage of this determination is that the proteins are simultaneously obtained by precipitation with the extraction of metabolites.

Organic solvents are commonly employed for metabolite extraction due to their ability to disrupt cellular membranes and solubilize a wide range of metabolites. However, the choice of solvent can significantly influence the efficiency and selectivity of metabolite extraction, ultimately impacting the reliability and interpretability of metabolomics data. Extraction techniques play a pivotal role in this process, influencing the quality and depth of the metabolite coverage obtained [7,8,11,12]. Non-aqueous organic solvents are determined to be the best choice when extracting primarily non-polar components. Andresen et al., 2022 [12] recommend 100% isopropanol, while Fritzche-Guenther et al., 2022 [34] obtained the best results with 100% ethanol extraction for all sets of metabolites. Otherwise, comparing the extraction efficiencies of 100% Et-OH, Et-OH-P, and 100% MetOH shows that similar numbers of polar metabolites are obtained, specifically amino acids, biogenic amines, and hexoses. Dettmer et al. [7] showed that pure acetone yielded the lowest extraction efficiency while methanol, methanol/water, methanol/isopropanol/water, and acid– base methanol recovered similar metabolite amounts. With respect to the polar metabolites analyzed by NMR, we chose aqueous reagents with different polarities for our comparison study. Our experiments evaluated methanol, ethanol, acetonitrile, an MTBE–methanol mixture, and methanol–chloroform. Chloroform and MTBE are a non-polar solvents that are particularly effective in extracting lipids and other hydrophobic metabolites, but as our results show, methods that use an MTBE–methanol mixture are also suitable for the extraction of polar metabolites, which is consistent with the observation of Luo et al., 2020 [19]. In addition to these individual solvents, mixtures or combinations of solvents can be used to improve the efficiency of metabolite extraction [12]. Such procedures were not included in our study. A certain inhomogeneity in the results of different working groups is also caused by the application of agents to different cell lines and different analytical techniques. In the case of adherent mammalian cells, many works are based on analyses that use carcinoma cell lines [10,19,26,35]. Therefore, we hope that optimization studies on the isolation of intracellular metabolites using mesenchymal stem cells and human dermal fibroblasts can contribute to the enrichment of optimization studies.

## 5. Study Limitation

One limitation of this study is the relatively small data set, which corresponds to the amount of metabolites identified using NMR. In addition to the choice of organic solvent and harvesting methods, other factors such as extraction time, temperature, and agitation also influence the efficiency of metabolite extraction. Optimization of these parameters is also crucial to ensure an accurate representation of the intracellular metabolome, but they were not the subject of our study. Another limitation may be the difference in protein precipitation when different organic agents are used.

## 6. Conclusions

The comparative analysis of extraction methods for intracellular metabolomics underscores the importance of selecting appropriate techniques to achieve optimal metabolite isolation. By considering the advantages and limitations of various approaches, researchers can tailor extraction protocols to their specific research objectives, advancing our understanding of cellular metabolism and disease mechanisms through comprehensive metabolomics profiling.

In conclusion:Direct scraping to organic solvent is a method that yields higher abundances of most determined metabolites;The comparison of scraping and trypsinization confirmed statistically significant differences in several metabolites, mainly in the classes of amino acids and peptides, for both types of cells;Applying different temperatures of washing solution before direct scraping into the organic agent, as well as comparing of different trypsinizing solutions, did not show statistically significant differences;The observed % CV depended on the cells and the harvesting or extraction methods;The comparison of several extraction methods showed statistically significant differences in several metabolites, mainly in the classes of amino acids and peptides and then in purine and pyrimidine nucleotides, their sugars, and derivatives in favor of the MTBE method;Extractions with the use of different methanol and acetonitrile polar reagents showed mostly the same quality;For both the HDFa and DPSC cells, the MTBE method, methanol–chloroform, and 80% ethanol extractions showed higher extraction efficiencies for the most identified and quantified metabolites.

## Figures and Tables

**Figure 1 metabolites-14-00268-f001:**
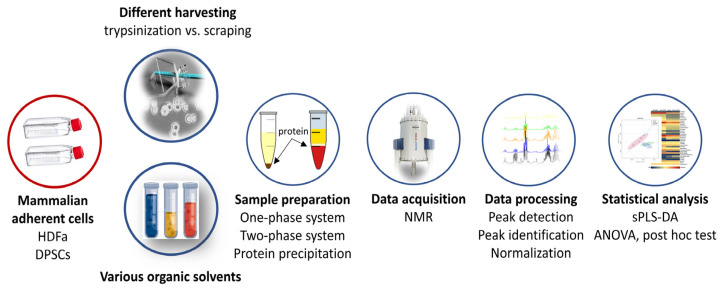
Experimental design—analytical workflow.

**Figure 2 metabolites-14-00268-f002:**
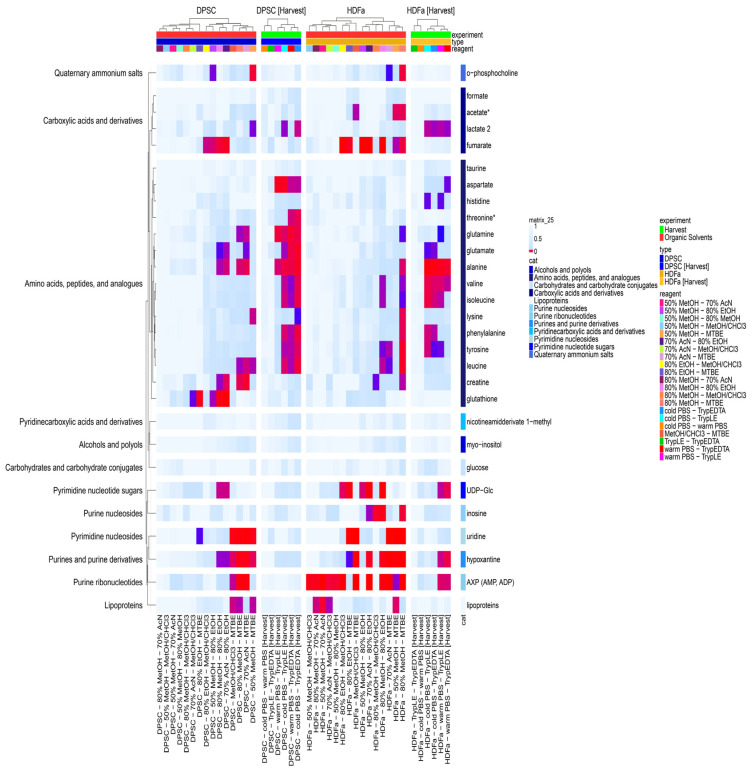
Heat map showing modulation of 29 identified and quantified metabolites (according to statistical significance: ANOVA *p*-value followed by Tukey’s post hoc test with Benjamini–Hochberger correction, corrected *p*-value <0.05) when assessing the impact of different harvesting methods or extraction with different organic agents. * peaks of metabolites were very slightly overlapping with other ones.

**Figure 3 metabolites-14-00268-f003:**
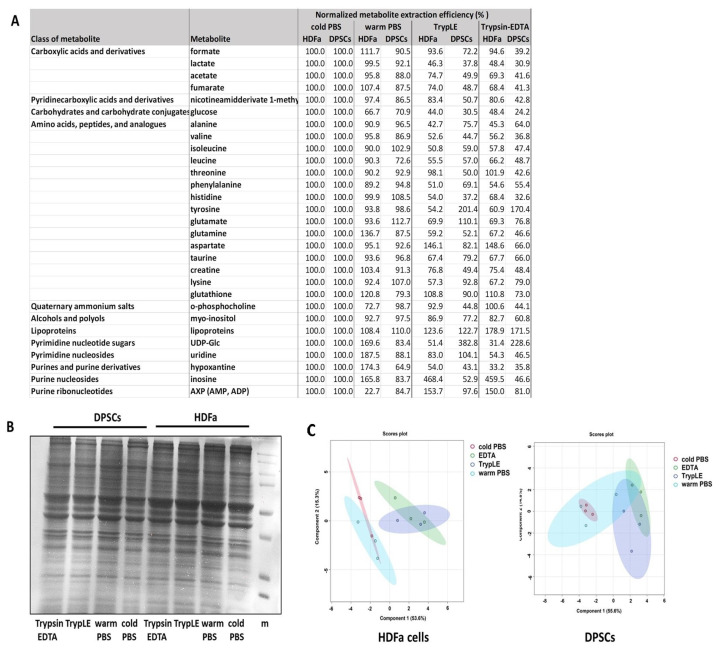
Effect of different harvesting methods on the isolation of metabolites. The cells directly scraped into 50% methanol after washing with cold PBS were determined as a reference set. Data are given as mean (in %) (**A**). A total of 30 µg of protein lysate was loaded on a representative 12% PAGE gel (**B**) to control protein quality. sPSL-DA plot. sPLS-DA visualization of different harvesting methods; normalized levels of metabolites were used as input variables (**C**).

**Figure 4 metabolites-14-00268-f004:**
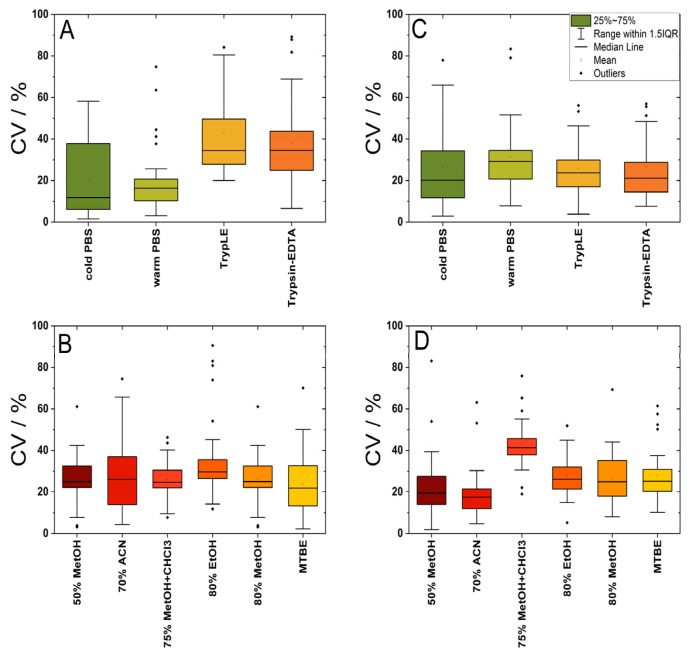
Effect of different harvesting and extraction methods on the average correlation coefficient (CV/%). The HDFa cells were directly scraped or trypsinized (**A**) and extracted into different solvents (**B**). DPSC cells were directly scraped or trypsinized (**C**) and extracted into different solvents (**D**).

**Figure 5 metabolites-14-00268-f005:**
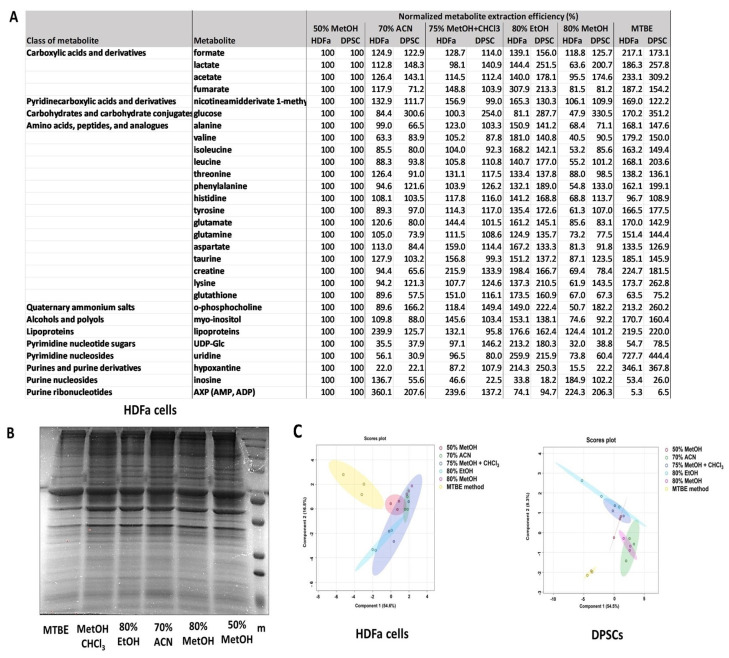
The effect of various organic solvents on metabolite isolation. Cells directly scraped into 50% methanol after washing with cold PBS were determined as a reference set. Data are given as mean (in %) (**A**). Control of protein quality. A total of 25 µg protein lysate was loaded on a representative 12% PAGE gel (**B**) sPSL-DA plot. sPLS-DA visualization of different harvesting methods; normalized levels of metabolites were used as input variables (**C**).

## Data Availability

All data generated or analyzed during this study are included in the main text of the article and its electronic Appendix A. Raw NMR spectra as well as data evaluated are available upon request from Eva Baranovičová.

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
