# Peer review of "Comparison of Various Extraction Approaches for Optimized Preparation of Intracellular Metabolites from Human Mesenchymal Stem Cells and Fibroblasts for NMR-Based Study"

_metabolites, 2024, doi:10.3390/metabo14050268_

Round 1
Reviewer 1 Report
Comments and Suggestions for Authors
The article submitted by Nováková et al is devoted to investigation of different sample treatment protocols intended to preparation of cells samples for metabolomics screening by NMR. The study is useful for investigators dealing with cell metabolomics; it is well designed and described. I have several remarks and questions which should be addressed before making my final opinion.
1) Page 2, line 60: Please explain the abbreviation ISCT here, where it is met for the first time, or just don't use it at all.
2) Subsections in the section 2 "Materials and methods" should be numbered.
3) Page 3, line 102: The index 2 in "CO2" should be subscripted.
3) What was the total number of cells taken for preparation of a single sample used for the metabolite extraction? This is my most important question.
4) Page 4, line 164: The scan type NOESY should be typed as the uppercase.
5) Page 4, line 168, 174: Shouldn't it be read as COSY?
6) Page 5, line 210: Unfortunately, no supplementary files were submitted by the authors. Also, please provide all raw data used for heatmap building and statistical analysis.
7) Page 10, line 371: I cannot agree with the authors' statement that MTBE is a solvent suitable for polar metabolites extraction which is based on the data from [19]. Luo et al, the authors if the study [19], developed a cell extraction approach protocol consisting of cell treatment with MTBE followed by water addition, centrifugation and evaporation of a part of the extract. This allows extraction of polar metabolites into water and lipids into MTBE. The sentence used in the manuscript under review implies that polar metabolites can be extracted with MTBE which is not possible actually, and I recommend to revise the text.
Author Response
The article submitted by Nováková et al is devoted to investigation of different sample treatment protocols intended to preparation of cells samples for metabolomics screening by NMR. The study is useful for investigators dealing with cell metabolomics; it is well designed and described. I have several remarks and questions which should be addressed before making my final opinion.
1.Page 2, line 60: Please explain the abbreviation ISCT here, where it is met for the first time, or just don't use it at all.
Information was added to text, line 54-55: ISCT (International Society for Cell and Gene Therapy)
2) Subsections in the section 2 "Materials and methods" should be numbered.
Thank you fo recommendation, subsections were numbered.
3) Page 3, line 102: The index 2 in "CO2" should be subscripted.
We corrected.
3) What was the total number of cells taken for preparation of a single sample used for the metabolite extraction? This is my most important question.
When we scraped the cells from flask direct to organic solvent, we didn´t have possibilities to count the cells. Use cells from different flask cultivated in parralel is not sufficient, because on different flask, cells can grown a bit differently. To this reason we normalized results to proteins which were precipitated continuosily during extraction and which reflect number of cells. Then, metabolite abundances were normalized to total protein amount.
4) Page 4, line 164: The scan type NOESY should be typed as the uppercase.
We corrected.
5) Page 4, line 168, 174: Shouldn't it be read as COSY?
We corrected.
6) Page 5, line 210: Unfortunately, no supplementary files were submitted by the authors. Also, please provide all raw data used for heatmap building and statistical analysis. √
All data were submitted, include chemical shifts (Supplementary table S1). Each from supplementary tables contains raw data, normalized data to total protein amount, Anova and post-hoc p-value used for Heat map preparation for certain cell and method (Supplementary tables S2-S5).
7) Page 10, line 371: I cannot agree with the authors' statement that MTBE is a solvent suitable for polar metabolites extraction which is based on the data from [19]. Luo et al, the authors if the study [19], developed a cell extraction approach protocol consisting of cell treatment with MTBE followed by water addition, centrifugation and evaporation of a part of the extract. This allows extraction of polar metabolites into water and lipids into MTBE. The sentence used in the manuscript under review implies that polar metabolites can be extracted with MTBE which is not possible actually, and I recommend to revise the text.
We corrected required sentence.
Line 392-395: Chloroform and MTBE are a nonpolar solvents that is particularly effective in extracting lipids and other hydrophobic metabolites, but as our results showed, method that using MTBE-methanol mixture is also suitable for polar metabolite extraction which is consistent with...........
We also added information about coeficient of variation, %CV. Lines: 266-281 and lines:311-321.
Reviewer 2 Report
Comments and Suggestions for Authors
The manuscript is quite interesting as metabolite extraction and sample preparation are the essential components of any Omics technology. My queries are below:
1. What is MTBE in the abstract?
2. Write down the conclusion in a precise manner in the abstract then the advantage to the scientific community as already done by authors.
3. There are a few methods developed and explored by a few groups that can be compared for the robustness of those methods for metabolite extraction for a wide range of polarities. Cite related references in the proper places. Only one set of data is sufficient to demonstrate the robustness of these methods.
4. Increase the resolution of figures.
5. Make a figure for stacked plot of each spectrum for the extraction method.
Author Response
The manuscript is quite interesting as metabolite extraction and sample preparation are the essential components of any Omics technology. My queries are below:
1.What is MTBE in the abstract?
MTBE is methyl tert-butyl ether. We added information to the text and make sentence more correct -MTBE method we replace to MTBE solvent.
- Write down the conclusion in a precise manner in the abstract then the advantage to the scientific community as already done by authors.
Thank you for comment. We supplemented the abstract.
- There are a few methods developed and explored by a few groups that can be compared for the robustness of those methods for metabolite extraction for a wide range of polarities. Cite related references in the proper places. Only one set of data is sufficient to demonstrate the robustness of these methods.
Thank you for the recommendation. The discussion on the robustness of the methods was expanded. Lines: 388-392
- Increase the resolution of figures.
We prepared.
- Make a figure for stacked plot of each spectrum for the extraction method
We prepared.
We also added information about coeficient of variation, %CV. Lines: 266-281 and lines:311-321.
Round 2
Reviewer 1 Report
Comments and Suggestions for Authors
The authors have revised their manuscript in accordance with all comments and questions, therefore I suggest acceptance of the submission.
Author Response
Thank you very much
Reviewer 2 Report
Comments and Suggestions for Authors
The authors have responded to all of the comments nicely. I recommend this to be published in this journal.
Author Response
Thank you very much